# Muscle Quality of Knee Extensors Based on Several Types of Force in Multiple Sclerosis Patients with Varying Degrees of Disability

**DOI:** 10.3390/medicina58020316

**Published:** 2022-02-19

**Authors:** Kora Portilla-Cueto, Carlos Medina-Pérez, Ena Monserrat Romero-Pérez, Gabriel Núñez-Othón, Mario A. Horta-Gim, José Antonio de Paz

**Affiliations:** 1Institute of Biomedicine (IBIOMED), University of León, 24071 León, Spain; cora1995port@gmail.com; 2Division of Biological Sciences and Health, University of Sonora, Hermosillo 83000, Mexico; ena.romero@unison.mx (E.M.R.-P.); gabriel@guaymas.uson.mx (G.N.-O.); mario.horta@unison.mx (M.A.H.-G.)

**Keywords:** muscle quality, multiple sclerosis, EDSS, neurological disability, DXA

## Abstract

*Background and Objectives*: Multiple sclerosis (MS) tends to affect muscle performance, mainly in the lower extremities. The degree of disability is associated with the loss of strength and muscle mass, to varying extents. Muscle quality (MQ) expresses the amount of force produced relative to the activated muscle mass. The purpose of this study was to compare the MQ of the knee extensors in the main manifestations of strength (isometric, dynamic strength, and power) among patients with differing degrees of neurological disability and evolutionary forms of the disease. We also establish reference values for MQ in MS patients (pwMS). *Materials and Methods*: In total, 250 pwMS were evaluated according to the Expanded Disability Status Scale (EDSS). The maximum dynamic and isometric forces and muscle power manifested a load of 60% of the maximum dynamics of the knee extensors. The lean mass of the thigh and hip was determined by densitometry, and the MQ was calculated for the three types of force evaluated. *Results:* The pwMS with relapsing remitting MS (RRMS) presented isometric MQ values that were 15.8% better than those of pwMS with primary progressive MS (PPMS) and 13.8% better than those of pwMS with secondary progressive MS (SPMS). For pwMS with SPMS, the dynamic MQ was 16.7% worse than that of patients with RRMS, while the power MQ was 29.5% worse. By degree of disability (<4 >7.5 EDSS score), patients with better MQ had mild EDSS scores, and patients with severe EDSS scores had 24.8%, 25.9%, and 40.3% worse isometric, dynamic, and power MQ scores, respectively, than those with RRMS. Based on these results, reference values for MQ in pwMS were established. *Conclusions:* The pwMS with different types of MS do not show differences in lean mass or strength but do show differences in MQ. In pwMS with different EDSS grades, there are no differences in lean mass, but there are differences in strength based on MQ, especially power MQ.

## 1. Introduction

Multiple sclerosis (MS) is a chronic, inflammatory, and immune-mediated [1] central nervous system (CNS) disease that results in the destruction of the myelin sheath of neuronal axons and can even lead to neurodegeneration [2]. MS is the most prevalent chronic inflammatory disease of the CNS [3], with an incidence three times higher in women than in men [4]. The clinical manifestations of MS can vary, with complex symptoms in the sphere of sensitivity, proprioception, balance, vision, mobility, muscle performance, sphincter control, and cognition [5]. Exacerbations occur frequently over time, followed by partial or total functional recovery (relapsing–remitting MS, RRMS), although MS can also present with a progressive accumulation of physical and cognitive deficits after a period of exacerbations (secondary progressive, SPMS) or without a prior phase of exacerbations (primary progressive, PPMS). In any of the evolutionary forms of MS, the accumulation of these deficits often leads to a progressive reduction in quality of life and deterioration of functional capacity [6].

From a physiological point of view, the manifestation of muscular force is the result of tension produced from contraction by the sarcomeres of the muscle cells. The magnitude of the tension generated by the contraction of a muscle or group of muscles is related, in part, to the type of motor units recruited, the amount of existing muscle mass, and the length and arrangement of muscle fascicles [7]. Muscle strength and size are significantly correlated (correlation adjusted for age and sex with a value of r = 0.26 [8]), but the magnitude of the correlation between muscle size and strength varies depending on the muscle group analyzed [9] and the type of strength evaluated (muscular power, maximum dynamic strength, and maximum isometric strength) [10,11,12]. The loss of muscle mass is associated with a loss of strength and functionality (e.g., in healthy elderly [13] and hemodialysis patients) [14]. However, the degree of strength loss is higher than the degree of muscle mass loss [15]. On the other hand, several studies found that the degree of disability is more strongly associated with a loss of strength than of muscle mass [16], but also noted an association between the degree of functionality in physical activities related to daily life (e.g., walking speed or getting up and sitting in a chair) and the degree of muscle quality (MQ), expressed as the manifested strength divided by the activated muscle mass [17].

Muscular performance (isometric strength, dynamic strength, and muscular power) is frequently affected in MS patients, especially in the lower extremities [18,19]. Factors such as gender, age, type of MS, and degree of neurological disability assessed according to the Expanded Disability Status Scale (EDSS) [20] have also been related to the ability to produce force, and some reference values are currently available for different manifestations of force among MS patients (men and women) with different types and degrees of neurological disability [21]. Compared to healthy individuals, patients with MS present a loss of muscle mass, a lower pennation angle, thinner muscle fibers on the basis of diameter, and a higher proportion of type II fibers, as well as the reduced ability to fully activate motor units and a reduced rate of force development [18,22,23,24]. The lower muscular performance of MS patients is accentuated in dynamic contraction exercises performed at high speed [25,26]. Muscle power (load × displacement/time) is a strong predictor of the degree of functional disability [27], which is why interest has grown in analyzing the effects of physical exercise programs based on muscular power work (exercises developed at high execution speeds) of the lower limbs (MMII) [28]. For these reasons, physical exercise is widely recommended for patients with MS [29,30]; in recent years, the importance of strength training has been especially emphasized among these patients [31].

The degree of functionality is not associated with muscle mass to the same extent as with muscle strength; indeed, throughout a patient’s functional decline, mass loss does not follow the same trajectory as strength loss [32]. These observations have contributed to increased interest in the study of MQ in recent years, especially in the context of aging, sarcopenia, or disability [33,34,35,36,37].

The concept of muscle quality (MQ) is interesting, as MQ provides an index for the level of functionality of muscle tissue relative to its mass. For example, this index was studied in relation to the ability to use glucose, oxidative damage, metabolism proteins, and muscle density [38]. However, publications that use the force production index in relation to muscle mass are more common [39]. This relationship is a key determinant of muscle function in adulthood [27]. MQ is considered a better marker of functional capacity than absolute strength or muscle mass [40]. One of the difficulties presented by the study of MQ is the lack of a widely accepted methodology for its evaluation. From the perspective of physical capacity, studies related to MQ have determined muscle mass (actually lean mass or fat-free mass) and cross-sectional muscle (CSA) using ultrasonography, bioimpedance analysis (BIA), dual-energy X-ray absorptiometry (DXA), computed tomography (CT), and magnetic resonance imaging (MRI) [41]. These studies have measured the strength manifested as isokinetic strength [42], isometric strength [43], or maximum strength (one rep maximum, 1 RM) [44]. Published scientific articles related to MQ have frequently used the elderly as study subjects, but not patients with MS. The interest shown in scientific publications on MQ research and, on the other hand, the lack of studies carried out on this topic in MS patients, highlight the need and relevance of such studies.

The purpose of this study was to compare the MQ of the knee extensors in the main manifestations of strength (isometric, maximum dynamic strength, and muscular power) between patients with differing degrees of neurological disability and evolutionary forms of the illness. We also provide reference values for muscle quality in patients with MS.

## 2. Materials and Methods

### 2.1. Sample

This study involved 250 pwMS (161 women and 89 men) of different ages. All patients belonging to MS patient associations in Castilla y León were offered participation in this research. We collected data related to the maximum strength of the leg extensors and body composition.

The inclusion criteria for the patients were a confirmed diagnosis of MS based on the McDonald criteria (2001) [45], the ability to walk with or without assistance, the ability to perform tests, and having performed a full-body DXA with an interval of less than one month from the strength assessment. In addition, patients for whom this type of effort was contraindicated by the responsible physician were excluded.

A doctor took the medical history of all participants in this investigation and assessed the degree of neurological disability using the EDSS scale [20]. The degree of neurological disability was categorized as mild = EDSS ≤ 2.5; moderate = EDSS ≤ 5; severe = EDSS ≤ 7.5; or very severe = EDSS > 7.5, following the criteria used previously in a similar study [21].

The methods, procedures, and data processing were in accordance with relevant ethical standards and the Declaration of Helsinki (revised in October 2013). Before the start of the study, patients signed their consent to participate in the study.

### 2.2. Measurement Methods

The maximum voluntary isometric contraction strength (MVIC), maximum dynamic strength (MDF), and displacement mean power (Pw) were bilaterally measured in the knee extensors using a multi-station machine (BH^®^ fitness Nevada Pro-T, Madrid, Spain), following the protocol used in previous studies [46,47]. This is detailed in the paragraphs below.

The length of the lever arm of the machine was adjusted according to the length of the lower leg so that when force was generated, that force was applied with the area of the tibial malleolus.

### 2.3. Isometric Force

The MVIC test was performed with a similar device described in previous works [46,47,48] and was measured using a load cell (Globus, Codogné, Italy; sampling frequency 1000 Hz). Data were collected and analyzed with the associated software (Globus Ergo Tester v1.5, Codognè, Italy). The participants sat upright, with a flexion angle at the hip joint of 110° and of approximately 90° at the knee, as measured using a TEC goniometer (Sport-Tec Physio & Fitness, Pirmasens, Germany). During the test, the patients were instructed to push as hard as possible from the start of the test and to maintain that maximal effort for five seconds. The participants performed two attempts, separated by a minute of rest. Only the highest peak value of the two was used for the analysis. All participants received verbal reinforcement while they exerted effort.

### 2.4. Maximum Dynamic Strength

For the maximum dynamic strength evaluation, we used the one repetition maximum (1 RM) protocol. This evaluation was performed ten minutes after the MVIC test, following previously published protocols [21,48]. First, four repetitions at 50% of the MVIC were performed. After each exercise, the patient indicated the subjective perception of his or her effort (RPE) through the OMNI-RES Endurance Exercise Scale [49].

Next, progressive series of two repetitions were performed until reaching a value of 1 RM in less than six attempts. Depending on the RPE and the quality of the technical execution of the last series performed, the load was increased by between 5 and 14 kg. The rest period between each series was two minutes. In the few cases (11) in which the 1 RM was not reached in a maximum of six attempts, the evaluation was repeated 48 h later. When the patient could not execute a second repetition, this mobilized load was considered the 1 RM, and in cases in which it was not possible to mobilize even once, an intermediate load was placed between the load that was mobilized both times and the load that could not be moved.

### 2.5. Muscle Power

The power test was performed with the same procedure used in previous work [47]. The mean power of the displacement was determined in five sets of three repetitions performed at maximum speed in the concentric phase, with two seconds of rest between each repetition. For the analysis, the highest value of the mean displacement power of the three repetitions of each series was taken. The loads used in the series were 40%, 50%, 60%, 70%, and 80% of the 1 RM. The order of the series was randomly administered for each subject [50]. The mean power was recorded using a rotational encoder (Globus Real Power; sample rate 300 Hz) with the associated software (Globus Real Power v3.11). Only a load that was 60% of the MVIC was taken for the analysis since, with this load, the patients showed a higher value of muscle power (Figure 1).

### 2.6. Muscle Mass

Muscle mass (lean mass) was evaluated using DXA, (Prodigy Primo, General Electric^®^ and enCore 2009^®^ software version 13.20.033) by performing a full-body scan of the dorsal decubitus while the participant was dressed only in underwear and had arranged their lower limbs with standard separation, corresponding approximately to hip width apart. This evaluation was performed following the standards of the International Society for Clinical Densitometry Official Positions [51] and previously used protocols [48]. Measurements of fat mass (FM), lean mass (MM), and bone mineral content (BMC) were collected, both from the whole body and from each of the extremities.

### 2.7. Lean Mass of the Region of Interest 

Using the images generated by DXA, the lean mass of each region of interest (ROI) was determined to calculate muscle quality. The ROI was considered the region featuring most of the agonist, synergist, and antagonist muscles involved in the extension of the knees sitting at a hip angle of 110°, for which limits were determined with the following anatomical references: the superior border is delimited by the line tangent to the upper back’s radiological shadow of the iliac crests; the lower border is defined by the line that passes through the femorotibial space; the inner edge is the line perpendicular to the upper edge that runs between the two extremities; and the external border is the line outside the radiological shadow of the existing limb between the upper and lower border and perpendicular to the internal border (Figure 2).

The lean mass expressed in the Results section is the sum of the right and left ROI.

### 2.8. Muscle Quality 

The MQs for isometric strength, dynamic strength, and power were obtained by dividing the values of each of these strength manifestations by the lean mass of the defined ROIs [52,53].

### 2.9. Statistical Analysis

The descriptive analysis of quantitative variables is shown with the mean and standard deviation (sd). To analyze the normality of the distribution, the Kolmogorov–Smirnov test was used. The description of qualitative variables is shown as the count and percentage proportion. The eventual association of sex with MS type or degree of disability was analyzed using Pearson’s chi-squared test. The comparison of quantitative variables between men and women was performed using Student’s *t*-test. The comparison of quantitative variables between groups according to type of MS or degree of disability was carried out independently via one-factor ANOVA, with a Bonferroni post-hoc test used where appropriate. Correlations were analyzed by means of Pearson’s correlation analysis. *p* < 0.05 was considered to indicate statistically significant differences. All statistical tests were performed using SPSS 25.0 for Windows (IBM Inc., Chicago, IL, USA, EEUU). The effect size was calculated using Cohen’s d (d) for mean comparison and partial eta squared (η^2^) when ANOVA was performed. The values that we obtained and are proposing for use as reference values are found in the Appendix A, provided with the cut-off values between quartiles (Q1_Q2; Q2_Q3; Q3_Q4).

## 3. Results

Given the higher prevalence of MS in women, and since gender was not a selection criterion for participation in the study, the proportion of women in the sample was higher (64%), as expected, but there were no differences between the sexes in terms of age, BMI, years of disease evolution since diagnosis, or EDSS value (Table 1).

Most of the patients in the sample presented RRMS (70%) or, less frequently, PPMS (13%). The degree of disability presented by the patients was mostly mild or moderate (77% overall), and only 3% had a very severe degree of illness (3%) (Table 2). An association was found between the type of MS and the degree of EDSS (χ^2^ (6, N = 250) = 23.721, *p* < 0.001), with RRMS presenting a lower degree of neurological disability and SPMS a higher degree of severity.

There was no association observed between sex and the evolutionary type of MS (χ^2^ (2, N = 250) = 4.520, *p* < 0.104) or between gender and EDSS grade (χ^2^ (3, N = 250) = 2.817, *p* < 0.421). For this reason, data in Table 2, Table 3, Table 4, Table 5, Table 6, Table 7 and Table 8 are not broken down by gender. In the Appendix A, the data of tables are disaggregated by gender.

The lean mass of the selected ROIs is shown in Table 3 by MS type and in Table 4 by EDSS level. Only a few (seven) patients in the sample presented very severe EDSS scores, so they were not considered for the comparison of the variables between groups of neurological disability. Lean mass was similar between patients with different evolutionary types of the disease, and it was also similar between patients with different degrees of EDSS.

Table 5 shows the descriptive values of the study variables grouped by MS type. The patients in the RRMS group were younger, and those in the PPMS group were the oldest. The strength values manifested by knee extension in the three types of strength studied were similar between all disease groups. In general, the muscle quality of the different manifestations of strength was observed to be similar between RRMS and PPMS patients and lower in the group of patients showing SPMS compared to RRMS patients. However, the effect sizes were found to be very small; the proportion of variance explained by the type of MS varied between only 5% and 7%.

Table 6 shows the descriptive values of the study variables grouped by EDSS grade. Patients with mild EDSS scores presented greater strength compared to patients with a moderate or severe degree of neurological disability. The degree of disability can explain between 13% and 17% of the variance in manifestations of strength. Moreover, patients with mild EDSS scores presented better muscle quality indices compared to patients with more severe neurological functional impairment and, in general, patients with a severe degree presented the worst values. Between 23% and 27% of variance in muscle quality is explained by the degree of the EDSS.

Table 7 and Table 8 summarize the threshold values used to establish the reference quartiles for the manifestations of muscle strength and quality depending on the type of MS and the level of neurological disability. The Appendix A provide the reference values grouped by sex, type of MS, and EDSS.

## 4. Discussion

After analyzing muscle quality in a wide sample of 250 patients with confirmed MS, differences in muscle quality were found to be greater when patients were compared according to the degree of neurological disability than when compared according to the disease evolutionary type. In total, only 36% of the sample was male because the incidence of this disease is three times higher in women than in men [54]. Overall, 70% of the sample had an RRMS course, which is explained by the fact that more than 80% of MS patients begin with an RR course, and only a minority have PPMS when diagnosed [55].

The average age in the sample was around 47 years—that is, a young adult population. However, since MS is a chronic disease with a tendency to progress, the time elapsed since the initial diagnosis of the disease was more important [56]. In the studied sample, the average time since diagnosis was around 11 years. As the sample was young, it is clear that MS is often diagnosed in a very young population [57].

The degree of neurological disability is the most widely used clinical parameter to monitor disease progression [56], and the most commonly used instrument for evaluating disability in clinical trials is the EDSS [58,59], which provides a score that ranges from 0 (no apparent neurological disability) to 10 (death from MS). However, there are no universally accepted cut-off points to categorize the degree of neurological disability based on the score obtained, which is why some authors use their own categories [30]. Based on the categories used in our center, 77% of the sample had a mild or moderate neurological disability. In total, 3% of our sample had a very severe degree of disability. This group of patients was not used in the present study for comparisons between patients based on disability because very few patients had a severe disability, especially considering the sizes of the other disability groups. Thus, comparisons with this group would have very poor statistical power. In our sample, the degree of the EDSS was associated with the evolutionary disease type, with RRMS having a higher proportion of patients presenting mild disability and patients with SPMS having a higher proportion of severe disability. Although our study showed a small correlation between age and EDSS score (Pearson’s *r* = 0.267), and the RRMS group was younger, these factors do not seem to provide an explanation for this difference in the proportion of disability since the correlation between the years of evolution of the disease and the EDSS score in our study was slightly higher (Pearson’s *r* = 0.323), and the time of evolution between the groups was similar. Our results agree with previous longitudinal studies which found that patients with RRMS tended to present with a slower progression of neurological disability [60]. One of the criticisms made of the universally used EDSS is that some important manifestations of MS patients, such as fatigue, upper extremity functions, and cognitive impairment, have little weight in the finally obtained score; thus, this instrument is not very sensitive to change [61]. The differences that we found in the EDSS between the groups were likely due to different affectations of nonmotor functions explored with the EDSS (vision, brainstem, sphincter control, or skin sensitivity), since we did not find differences between the groups in maximum isometric force, neither in dynamic force nor in the mean power manifested at 60% of 1 RM.

Some authors have studied lean mass in people with MS because lean mass is considered a key concept in force production [62]. However, these studies often focus on comparisons between patients with MS and a healthy population [63,64]. Very few studies have compared the lean mass determined by DXA between patients with different EDSS scores [30]. In our study, there were no differences between the type of MS and the lean mass of the pelvic–thigh area. There were also no differences observed between the different groups of degree of disability, unlike the results of another study [30], which observed that patients with a greater degree of disability presented a lower lean mass. However, this previous study determined the total lean body mass and only grouped patients into two disability groups (26 patients with EDSS scores less than 4 versus 21 patients with scores of more than 4.5). Moreover, the group with the greatest disability had about 17 years of disease evolution compared to 9 years of evolution in the group with the lowest disability.

The muscle quality of patients with MS, understood as the manifested force relativized to the activated muscle mass, is an aspect little explored in the scientific literature. We collected data on the ROIs defined by DXA in studies related to muscle quality of the knee extensors in different populations without MS. Some studies determined the lean mass of the entire limb [65,66,67], while others focused on the thigh region [68,69,70]. The ROIs used in our study were limited to obtain the majority of the lean mass of the main muscle groups that intervene as agonists, synergists, antagonists, or fixators in the extension movement of the knee, with an angulation of 110° at the hip [71].

The results of the analysis of muscle quality in the three evolutionary types of disease show that the MQ_Isometric was significantly higher in the group with RRMS (15.8% lower in the PPMS and 13.4% in the SPMS), although the effect size was relatively small, explaining 5% of the variance by evolutionary type of the disease. For the MQ_MDF, the RRMS group also presented significantly better quality than the SPMS group, which provided 17.7% worse quality and was not significant compared to the PPMS group, explaining 6% of the variability by evolutionary group of the disease. The difference in muscle quality was greatest in MQ_Power, which was 29.5% better in the RRMS group than in the SPMS group. Although this result was not significant, it was 16.1% better than the result for PPMS, explaining 7% of variance due to the evolutionary type of the disease. This greater difference is likely because a power exercise is neuromuscularly more complex, requiring more rapid interaction of agonists, synergists, and antagonists [72], and neural involvement has a greater impact on more complex exercises. Our study was cross-sectional, but considering that patients with SPMS of an earlier stage presented RRMS [6], our results suggest that the evolution of RRMS to SPMS tends to worsen neurological disability and muscle quality, despite not being accompanied by a loss of strength or muscle mass. 

When muscle quality was compared between MS patients with different degrees of neurological disability, the differences were clearer than those for the evolutionary type of the disease. Patients with a mild degree of EDSS scores presented significantly higher MQ_Isometric values than patients with moderate or severe involvement (11% and 24.8% better, respectively) and, in turn, severe EDSS scores were worse than those of moderate EDSS scores. The degree of disability ultimately explained 27.1% of the variance in MQ_Isometric. The group with a mild EDSS grade also presented significantly better muscle quality for the manifestation of maximum dynamic force—14.1% better than the moderate EDSS group and 26.9% better than the severe EDSS group. These differences were even greater in relation to MQ_Power, for which the quality in the mild EDSS group was 26.7% better than that in the moderate impairment group, and 40.3% better than that in the group with a severe degree of neurological disability. Overall, there were no differences in lean mass between the groups with different neurological disabilities, but there were differences in manifested strength, which confirmed that decreased muscle quality was caused by a lower capacity to generate force per unit of muscular mass. Further studies are needed to identify whether the observed loss in force generation is caused by changes in the muscle fibers or by changes in the neural components of muscle contraction.

Muscle quality in patients with MS has rarely been studied. To fill this gap, the present study provides values that can be used as a reference. These values were obtained using reproducible and sensitive methods to determine the segmental muscle mass of the knee extensors by applying DXA and measuring the main manifestations of force involved in the actions of daily life, including isometric force, dynamic force, and muscular power. In addition, the values were obtained from a large sample of patients and categorized according to both the evolutionary type of the disease and the degree of neurological disability. These muscle quality values are also provided in the Appendix A, where they are further disaggregated by sex.

However, the present study has some limitations. The main limitation is that this study utilized a cross-sectional approach. Although it can help establish hypotheses, this approach cannot elucidate the causes of observed differences. These causes should be investigated in future observational or longitudinal studies. Moreover, the determination of segmental lean mass through DXA, despite its excellent repeatability and reproducibility, does not allow for separate analysis of each muscle group or muscle.

## 5. Conclusions

In this study, pwMS with different types of MS did not show differences in lean mass or in manifestations of strength. However, patients did present differences in MQs, which were worse in patients with SPMS, especially for the MQ related to muscle power. The pwMS with different degrees of the EDSS also did not present differences in lean mass but did present differences in strength and MQ. Patients with mild degrees of the EDSS showed better MQs, especially for the MQ related to muscle power.

Reference values for the MQs are available as Appendix A, (https://www.mdpi.com/article/10.3390/medicina58020316/s1. Forces and MQ References Values by sex, EDSS level and Type Multiple Sclerosis).

## Figures and Tables

**Figure 1 medicina-58-00316-f001:**
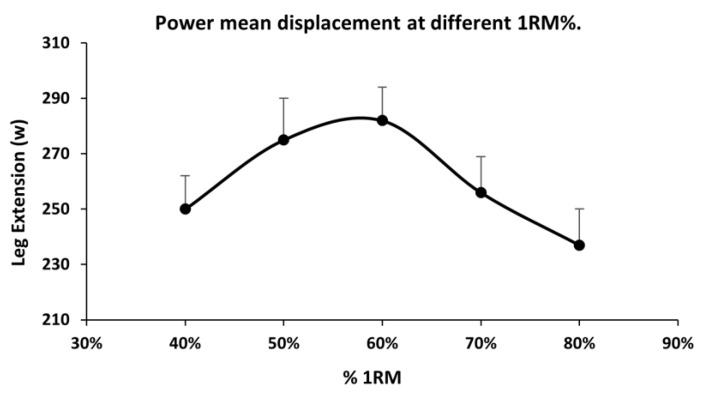
Mean displacement power at different percentages of 1 RM. w: wat; 1 RM: one repetition maximum; %: load percentage.

**Figure 2 medicina-58-00316-f002:**
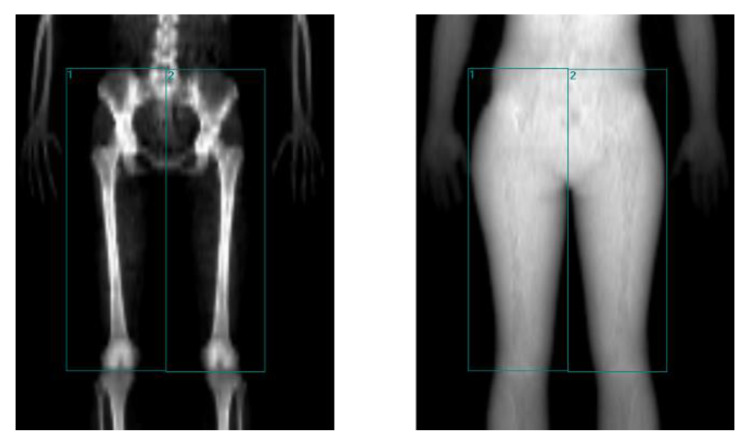
Schematic of the different ROIs used to estimate thigh lean mass by DXA.

**Table 1 medicina-58-00316-t001:** Characteristics of the sample population according to gender.

	Male (*n* = 89)			Female (*n* = 161)				
	Mean ± SD	Max	Min	Mean ± SD	Max	Min	*p*	*d*
Age (years)	45.2 ± 11.5	77	28	46.5 ± 11.5	75	20	0.433	0.113
BMI (kg/m^2^)	24.7 ± 4.2	37.9	9.7	24.2 ± 4.4	39.3	16	0.384	0.116
Years since diagnosis	10.8 ± 8.8	41	0	11.0 ± 7.5	30	0	0.934	0.025
EDSS	3.7 ± 2.1	8	1	3.4 ± 2.1	8.5	1	0.224	0.143

Max, Min: highest and lowest value, respectively, of the subsamples; BMI: body mass index; EDSS: Expanded Disability Status Scale; *p*: *p* value; *d*: Cohen’s *d* effect size.

**Table 2 medicina-58-00316-t002:** Distribution of the sample according to the type of MS and neurological disability level.

		MS Type			
		RR (%)	PP (%)	SP (%)	∑ (%)
EDSS level	Mild	82 (46)	8 (25)	3 (7)	93 (37)
	Moderate	65 (37)	14 (44)	20 (48)	99 (40)
	Severe	24 (14)	9 (28)	18 (43)	51 (20)
	Very Sev	5 (3)	1 (3)	1 (2)	7 (3)
	∑ (%)	176 (70)	32 (13)	42 (17)	250

∑: total MS type or grade of EDSS. RR: relapsing–remitting; PP: primary progressive; SP: secondary progressive.

**Table 3 medicina-58-00316-t003:** Lean mass (kg) of the ROIs as a function of EM type.

		RR	PP	SP		
		Mean ± SD	Mean ± SD	Mean ± SD	*p*	*η*²
ROI (kg)	Thighs	16 ± 3.6	17.3 ± 4.1	15.6 ± 3.7	0.079	0.021
	Right	7.8 ± 1.9	8.7 ± 2.4	7.8 ± 1.9	0.076	0.022
	Left	7.7 ± 1.8	8.5 ± 2.1	7.8 ± 1.9	0.121	0.018

ROI: region of interest; thighs: bilateral; ROI right: right thigh; ROI left: left thigh; RR: relapsing–remitting; PP: primary progressive; SP: secondary progressive; *p: p* value ANOVA; *η*²: partial eta squared size effect.

**Table 4 medicina-58-00316-t004:** Lean mass (kg) of the ROI as a function of neurological disability level by EDSS.

		EDSS				
		Mild	Moderate	Severe		
		Mean ± SD	Mean ± SD	Mean ± SD	*p*	*η*²
ROI (Kg)	Thighs	16 ± 3.6	15.3 ± 3.7	15.8 ± 3.8	0.405	0.009
	Right	8.1 ± 0.1	7.6 ± 2.1	7.8 ± 1.9	0.409	0.009
	Left	8.0 ± 1.7	7.6 ± 1.8	7.9 ± 2.0	0.438	0.008

EDSS: Expanded Disability Status Scale; ROI: region of interest; thighs: bilateral; ROI right: right thigh; ROI left: left thigh; *p: p* value ANOVA; *η*²: partial eta squared size effect.

**Table 5 medicina-58-00316-t005:** Muscle strength and quality values according to the type of MS.

MS Type													
	RR (R)			PP (P)				SP (S)					
	Mean ± SD	Max	Min	Mean ± SD	Max	Min	%	Mean ± SD	Max	Min	%	*p*	*η*²
Age (years)	42.4 ± 10 ^SP^	68.0	20.0	55.9 ± 6.6 ^S^	72.0	43.0		53.3 ± 12.6	77.0	31.0		0.000	0.234
Years since diagnosis	11.0 ± 7.7	41.0	0.0	7.9 ± 7.6	28.0	0.0		12.6 ± 8.9	31.0	0.0		0.086	0.002
MVIC (kg)	84.4 ± 32.6	194.1	20.0	78.3 ± 33.6	163.4	27.9	92.8	72.4 ± 28.8	144.7	21.0	85.8	0.094	0.028
1 RM (kg)	75.1 ± 29.0	150.0	6.0	74.8 ± 32.2	120.0	14.0	99.6	64.2 ± 26.9	130.0	12.0	85.5	0.131	0.020
Mean Power (w)	285.2 ± 146.7	917.2	51.0	269.3 ± 149.5	554.0	31.0	94.4	210.5 ± 127.2	576.0	58.0	73.8	0.095	0.033
MQ_ Isometric	5.4 ± 1.5 ^SP^	13.2	1.6	4.5 ± 1.3	7.4	2.0	84.2	4.7 ± 1.6	10.3	1.8	86.6	0.002	0.054
MQ_MDF	4.9 ± 1.5 ^S^	8.5	0.5	4.2 ± 1.4	6.9	1.2	86.0	4.0 ± 1.2	6.0	1.0	82.3	0.002	0.059
MQ_Power (w/kg)	18.0 ± 7.3 ^S^	53.4	3.6	15.1 ± 7.5	32.9	1.6	83.9	12.7 ± 5.6	26.4	4.6	70.5	0.004	0.076

MS: multiple sclerosis; RR: relapsing–remitting; PP: primary progressive; SP: secondary progressive; Max, Min: highest and lowest values, respectively, of the subsamples; MVIC: maximum voluntary isometric contraction; 1 RM: one rep maximum; Mean Power: average load displacement power; MQ_Isometric: muscle quality for isometric strength; MQ_MDF: muscle quality for dynamic strength; MQ_Power: muscle quality for average displacement power; *p: p* value ANOVA; *η*²: partial eta squared size effect; MVIC; 1 RM; Superscript letters indicate between-group differences from post-hoc analysis (P = PP; R = RR, S = SP); %: percentage value of the group mean variable with respect to the RR group measurement.

**Table 6 medicina-58-00316-t006:** Muscle strength and quality values according to the neurological disability level.

	EDSS Level												
	Mild (L)			Moderate (M)				Severe (S)					
	Mean ± SD	Max	Min	Mean ± SD	Max	Min	%	Mean ± SD	Max	Min	%	*p*	*η*²
Age (years)	41.6 ± 11.2 ^MS^	68.0	20.0	47.7 ± 9.3	66.0	28.0		51.9 ± 13.1	77.0	20.0		0.000	0.118
Years since diagnosis	8.6 ± 7.2 ^MS^	28.0	0.0	12.2 ± 7.7	30.0	0.0		12.4 ± 9.2	41.0	0.0		0.005	0.059
MVIC (kg)	92.0 ± 32.4 ^MS^	194.1	42.0	78.3 ± 31.4	176.1	33.0	85.1	68.1 ± 27.8	161.3	20.0	74.0	0	0.081
1 RM (kg)	83.4 ± 27.8 ^MS^	130.0	18.5	70.5 ± 28.2	130.0	31.0	84.5	62.5 ± 30.6	150.0	6.0	74.9	0.001	0.076
Mean Power (w)	338.3 ± 155.9 ^MS^	917.2	51.0	245.6 ± 124.6	637.0	58.0	72.6	208.9 ± 144.1	725.0	31.0	61.7	0	0.125
MQ_ Isometric	5.7 ± 1.3 ^MS^	8.6	3.0	5.1 ± 1.4 ^S^	10.3	2.5	89.0	4.3 ± 1.3	7.2	1.6	75.2	0.000	0.124
MQ_MDF	5.3 ± 1.5 ^MS^	8.5	1.0	4.6 ± 1.2 ^S^	7.9	2.6	85.9	3.9 ± 1.5	6.7	0.5	73.1	0.000	0.132
MQ_Power (w/kg)	20.9 ± 8.3 ^MS^	53.4	3.6	15.3 ± 5.0	24.7	5.5	73.3	12.5 ± 6.8	34.0	1.6	59.7	0	0.196

EDSS: Expanded Disability Status Scale; Max, Min: highest and lowest value, respectively, of the subsamples; MVIC: maximum voluntary isometric contraction; 1 RM: one rep maximum; Mean Power: average load displacement power; MQ_Isometric: muscle quality for isometric strength; MQ_MDF: muscle quality for dynamic strength; MQ_Power: muscle quality for average displacement power; *p: p* value ANOVA; *η*²: partial eta squared size effect; MVIC; 1 RM; Superscript letters indicate between-group differences from post-hoc analysis (L = Mild; M = Moderate; S = Severe); %: percentage value of the variable of the group mean with respect to the measurement of the mild group.

**Table 7 medicina-58-00316-t007:** Cut-off points for quartiles of the variables according to the type of MS.

	MS Type								
	RR			PP			SP		
	Q1_Q2	Q2_Q3	Q3_Q4	Q1_Q2	Q2_Q3	Q3_Q4	Q1_Q2	Q2_Q3	Q3_Q4
MVIC (Kg)	57.6	78.3	94.7	71.8	92.0	106.0	50.0	68.1	87.0
1 RM (Kg)	50.0	70.5	90.0	62.0	83.4	106.0	42.0	62.5	87.0
Mean Power (w)	185.3	256.5	364.5	136.0	247.0	385.0	146.0	180.0	233.2
MQ_ Isometric	4.5	5.2	6.3	3.7	4.6	5.2	3.7	4.5	5.8
MQ_MDF	4.0	5.0	5.8	3.7	4.7	5.1	3.2	4.1	5.1
MQ_Power (w/Kg)	14.1	17.2	21	10.4	16	19.0	8.9	12.4	17.0

Q_Q: cutoff values between quartiles; MS: multiple sclerosis; RR: relapsing–remitting;PP: primary progressive; SP: secondary progressive; Max, Min: highest and lowest value, respectively, of the subsamples; Diagnostic: years since diagnosis; MVIC: maximum voluntary isometric contraction; 1 RM: one rep maximum; Mean Power: average load displacement power; MQ_Isometric: muscle quality for isometric strength; MQ_MDF: muscle quality for dynamic strength; MQ_Power: muscle quality for average displacement power.

**Table 8 medicina-58-00316-t008:** Cut-off points for quartiles of the variables according to the neurological disability level.

	EDSS Level								
	Mild			Moderate			Severe		
	Q1_Q2	Q2_Q3	Q3_Q4	Q1_Q2	Q2_Q3	Q3_Q4	Q1_Q2	Q2_Q3	Q3_Q4
MVIC (kg)	57.6	81.3	105.8	61.4	83.6	99.0	58.1	73.8	89.0
1 RM (kg)	55.0	78.5	107.0	54.0	75.1	100.0	42.0	65.1	85.0
Mean Power (w)	236.0	330.0	405.0	169.0	201.5	302.0	114.0	186.0	250.0
MQ_ Isometric	4.9	5.7	6.6	4.2	4.9	6.0	3.8	4.4	5.2
MQ_MDF	4.7	5.2	6.1	3.5	4.6	5.4	2.9	4.1	5.1
MQ_Power (w/kg)	16.9	19.5	25	12.1	15	17.8	7.7	11.7	15.4

Q_Q: cutoff values between quartiles; EDSS: Expanded Disability Status Scale; Max, Min: highest and lowest value, respectively, of the subsamples; MVIC: maximum voluntary isometric contraction; 1 RM: one rep maximum; Mean Power: average load displacement power; MQ_Isometric: muscle quality for isometric strength; MQ_MDF: muscle quality for dynamic strength; MQ_Power: muscle quality for average displacement power.

## Data Availability

Additional information is available upon request from the senior author, japazf@unileon.es.

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
