# Peer review of "Muscle Quality of Knee Extensors Based on Several Types of Force in Multiple Sclerosis Patients with Varying Degrees of Disability"

_medicina, 2022, doi:10.3390/medicina58020316_

Round 1
Reviewer 1 Report
The information is very interesting for those of us whom are neuromotor clinicians and the disease process of MS is important. This clinical significance to the different types of MS as well as overall implications observational and prognostical. It is very nicely done to note muscle quality in patients with MS has been rarely studied, and again would try to continue to correlate to clinician relevance and patient personal significance.
Author Response
We appreciate your time and effort in reviewing this paper. We are pleased that you find it well constructed and written and of interest to neuromuscular clinicians and researchers.
Thank you
Reviewer 2 Report
Dear authors,
I congratulate you on your meticulous research.
Only two issues that can improve the manuscript I would like to cite:
- The title does not say the type of study performed and only naming the normality values is a bit scarce for your research.
- Line 128 gives the feeling that they do not provide all the information of the protocol by naming it first the reference and then the description. Make sure that it is fully described.
- Indicate whether the study has been approved by an ethics committee in the manuscript.
- The discussion is too long, try to shorten it. Twice you mention the limitation that the study is cross-sectional, you can improve it.
best wishes
Author Response
Dear Reviewer, we have found your criticism and recommendations positive and very constructive. We are grateful for the in-depth review that you have made of your corrections and the observations that you point out to us.
The title does not say the type of study performed and only naming the normality values is a bit scarce for your research.
Authors’ response: Thank you for this suggestion, and also following the similar suggestion of reviewer 3, we have removed from the title “Reference Values”. The title is now: this sentence as:
- Muscle Quality of Knee Extensors Based on Several Types of Force in Multiple Sclerosis Patients with Varying Degrees of Disability.
Line 128 gives the feeling that they do not provide all the information of the protocol by naming it first the reference and then the description. Make sure that it is fully described.
Authors’ response: the methodology for the evaluation of assessed forces, the determination of the lean mass in regions of interest and muscle quality calculation, are detailed after the introductory paragraph to the measurement methods, from line 135 to line 207. To increase clarity, we have left the paragraph, (lines 131-132), as follows:
- …following the protocol used in previous studies [46,47], and which is detailed in the paragraphs below.
Indicate whether the study has been approved by an ethics committee in the manuscript.
Authors’ response: In this Journal, there is a specific heading: “Institutional Review Board Statement”, in which we have noted: “The study was conducted in accordance with the Declaration of Helsinki and approved by the Ethics Committee of Universidad de León (January 30, 2017; study number 1835)”. (Lines 489-491)
The discussion is too long, try to shorten it. Twice you mention the limitation that the study is cross-sectional, you can improve
Authors’ response: we have reviewed the discussion with a new look, and we believe that we have pointed out the characteristics of the sample in order to contextualise our results so that they can be better interpreted. Unfortunately, we have not been able to identify superfluous paragraphs that could be eliminated to make the discussion a little shorter. Although the discussion is not short, we believe that it makes the article easier to contextualise and read for non-experts.
In order not to emphasise so much that the study is cross-sectional, we have deleted the following paragraph: cross-sectional of line 112, remaining:
- This study included the participation of 250 people….
And deleted the old line 428:
- However, this hypothesis can only be accepted or rejected by conducting future longitudinal studies
Reviewer 3 Report
Review Letters
Dear Authors
The title of this study seems to be consistent with the Medicina. I think this paper will be better if some minor and major points are corrected.
Minor points
Line 1: In the case of the title, except for prepositions and conjunctions, the first letter must be capitalized.
Line 4: It would be desirable to delete "Reference values" from the title.
Line 197: The title of the figure should appear at the bottom of the figure.
Line 214: All "p" presented in this paper are statistical symbols and should be converted to italics.
Line 322: (Ortona et al., 2016) should be replaced by reference number.
Major points
Introduction: The need for research in this study is very weak. At the bottom of the introduction, why is this study necessary and the purpose of the study should be clearly presented. That is, it is necessary to highlight the need for research.
Methods: The sample recruitment process should be described more specifically now. In addition, it is necessary to describe the selection criteria and exclusion criteria of the subjects.
It seems that the contents presented in your manuscript need to be corrected in English.
Sincerely,
Author Response
Dear Reviewer, we would like to thank you for your interest and time spent in evaluating and making comments and corrections to the manuscript, which undoubtedly improve our work. Thank you very much for your help.
Line 1: In the case of the title, except for prepositions and conjunctions, the first letter must be capitalized.
Authors’ response: thanks for the correction, it has been changed, and remains as follows:
- Muscle Quality of Knee Extensors Based on Several Types of Force in Multiple Sclerosis Patients with Varying Degrees of Disability.
Line 4: It would be desirable to delete "Reference values" from the title.
Authors’ response: as suggested by you, we have removed "reference values" from the title.
Line 197: The title of the figure should appear at the bottom of the figure.
Authors’ response: thanks for the correction, the title has been placed at the bottom of the figure.
Line 214: All "p" presented in this paper are statistical symbols and should be converted to italics.
Authors’ response: we have italicised all the p's in the text and tables. Thank you.
Line 322: (Ortona et al., 2016) should be replaced by reference number.
Authors’ response: thank you very much for pointing out this error, has been corrected and incorporated in the bibliographical references, renumbering all subsequent references.
Introduction: The need for research in this study is very weak. At the bottom of the introduction, why is this study necessary and the purpose of the study should be clearly presented. That is, it is necessary to highlight the need for research.
Authors’ response: At the reviewer's suggestion, and to underline the relevance of this study, we have added the following paragraph (lines 102-105):
- Published scientific articles related to MQ have frequently used the elderly as study subjects but not patients with MS. The interest shown in scientific publications on QM research and on the other hand the lack of studies carried out on this topic in MS patients, highlight the need and relevance of such studies.
Methods: The sample recruitment process should be described more specifically now.
Authors’ response: Participation in the study was offered to all patients belonging to the MS Associations of Castilla y León. To make it clearer, we have modified the paragraph to read as follows:
- This study involved 250 pwMS (161 women and 89 men) of different ages. All patients belonging to MS patient associations in Castilla y León were offered participation in this research. (lin 112-114)
In addition, it is necessary to describe the selection criteria and exclusion criteria of the subjects.
Authors’ response: The inclusion criteria are indicated. None of the participants were excluded as all were eligible to participate in the study. However, in response to your comment and in order to formally express it, we have added:
- In addition, patients for whom this type of effort was contraindicated by the responsible physician were excluded. (lin 119-120)
It seems that the contents presented in your manuscript need to be corrected in English.
Authors’ response: we have sent the article for English language correction to MDPI, (English Editing ID:english-39434). This service certifies that: “has undergone English language editing by MDPI. The text has been checked for correct use of grammar and common technical terms, and edited to a level suitable for reporting research in a scholarly journal. MDPI uses experienced, native English speaking editors”